# Decreased Expression of KLF4 Leading to Functional Deficit in Pediatric Patients with Intestinal Failure and Potential Therapeutic Strategy Using Decanoic Acid

**DOI:** 10.3390/nu15122660

**Published:** 2023-06-07

**Authors:** Junkai Yan, Yuling Zhao, Lu Jiang, Ying Wang, Wei Cai

**Affiliations:** 1Division of Pediatric Gastroenterology and Nutrition, Xinhua Hospital, School of Medicine, Shanghai Jiao Tong University, Shanghai 200092, China; yanjunkai@xinhuamed.com.cn; 2Shanghai Institute for Pediatric Research, Shanghai 200092, China; jianglu@xinhuamed.com.cn; 3Shanghai Key Laboratory of Pediatric Gastroenterology and Nutrition, Shanghai 200092, China; 4Department of Pediatric Surgery, Xinhua Hospital, School of Medicine, Shanghai Jiao Tong University, Shanghai 200092, China; yulingzyl@sjtu.edu.cn

**Keywords:** intestinal failure, parenteral nutrition, single-cell RNA sequencing, Kruppel-Like Factor 4, decanoic acid

## Abstract

Pediatric intestinal failure (IF) is the reduction in gut function to below the minimum necessary for the absorption of macronutrients and/or water and electrolytes, such that intravenous supplementation is required to maintain health and/or growth. The overall goal in treating IF is to achieve intestinal adaptation; however, the underlying mechanisms have not been fully understood. In this study, by performing single-cell RNA sequencing in pediatric IF patients, we found that decreased Kruppel-Like Factor 4 (KLF4) may serve as the hub gene responsible for the functional deficit in mature enterocytes in IF patients, leading to the downregulation of solute carrier (SLC) family transporters (e.g., SLC7A9) and, consequently, nutrient malabsorption. We also found that inducible KLF4 was highly sensitive to the loss of certain enteral nutrients: in a rodent model of total parenteral nutrition mimicking the deprivation of enteral nutrition, the expression of KLF4 dramatically decreased only at the tip of the villus and not at the bottom of crypts. By using IF patient-derived intestinal organoids and Caco-2 cells as in vitro models, we demonstrated that the supplementation of decanoic acid (DA) could significantly induce the expression of KLF4 along with SLC6A4 and SLC7A9, suggesting that DA may function as a potential therapeutic strategy to promote cell maturation and functional improvement. In summary, this study provides new insights into the mechanism of intestinal adaptation depending on KLF4, and proposed potential strategies for nutritional management using DA.

## 1. Introduction

Pediatric intestinal failure (IF) is the reduction in gut function to below the minimum necessary for the absorption of macronutrients and/or water and electrolytes, such that intravenous supplementation is required to maintain health and/or growth [1,2,3]. In addition to enteral nutrition (EN), some IF patients may remain dependent on parenteral nutrition (PN) for years or even total parenteral nutrition (TPN) if EN cannot be tolerated. The natural history of IF is variable and is largely influenced by underlying disorders. However, the most common types of IF in children may be categorized into three main groups: (1) anatomical disorders such as short bowel syndrome (SBS), (2) motility disorder such as chronic intestinal pseudo-obstruction (CIPO), and (3) mucosal intestinal diseases such as microvillous inclusion disease (MVID). In principle, one of the most important goals for treating IF is to achieve PN independence by promoting intestinal adaptation, which is a natural compensatory process that occurs in the remnant bowel and improves nutrient and fluid absorption [4]. Overall, intestinal adaptive processes include the accelerated proliferation of stem cells (SC), differentiation of enterocyte progenitors (proEC), and functional improvement of mature enterocytes (mEC) characterized by increased expression of transporters and nutrient/ion exchangers. However, the underlying molecular mechanisms, as well as the potential role of enteral nutrients, have not been fully understood.

Recently, a wide variety of single-cell RNA sequencing (scRNA-seq) methods have been developed that can accurately investigate the transcriptomes of mammalian cells at a single-cell resolution, which has greatly improved the elucidation of the transcriptome dynamics during the processes of cell proliferation, differentiation, and maturation [5,6]. Using scRNA-seq, breakthroughs concerning the development of gastrointestinal tract have emerged. For instance, scRNA-seq combined with laser capture microdissection of villi unraveled the functional zonation of enterocytes along the villus axis [7]. In addition, scRNA-seq combined with validation using organoid techniques revealed regional reprogramming during intestinal adaptation in a rodent model of massive small bowel resection [8]. However, transcriptomic regulations in human pediatric IF patients have not been systematically addressed at a single-cell resolution, limiting the discovery of novel therapeutic targets and the development of new strategies for promoting intestinal adaptation.

In this study using 10× genomic scRNA-seq, we found that Kruppel-Like Factor 4 (KLF4), a zinc finger transcription factor that has recently been identified as a pivotal modulator during the differentiation and maturation of enterocytes, decreased dramatically in the mature enterocytes of IF patients, along with a variety of downregulated solute carrier (SLC) family transporters [9,10]. Therefore, we hypothesized that KLF4 may play a role in the maturation program of the intestinal mucosa to achieve adaptation after bowel resection. Moreover, by using a rodent model of total parenteral nutrition (TPN), we demonstrated that KLF4 expression was highly dependent on the stimulation of enteral nutrients. Furthermore, by using IF patient-derived organoids (PDOs), we found that decanoic acid (DA) may serve as a potent nutritional inducer of KLF4. Our study provided novel insights into the mechanism of intestinal adaptation targeting KLF4, and may be of assistance for the development of potential strategies using DA.

## 2. Materials and Methods

### 2.1. Patients

Six pediatric patients undergoing small bowel anastomosis from January 2022 to March 2023 were enrolled in this study, including 3 IF patients and 3 disease controls. Two patients with ileal duplication and a patient with abdominal cyst were recruited as control, because these patients represented normal bowel function and had never received parenteral nutrition. Normal biopsies were taken when they were receiving anastomosis for minimal bowel resection, which was confirmed by histological examination. The diagnosis of IF was based on PN dependence as described previously [11]. Nutritional status was considered to be malnutrition when the patient’s weight-for-age was below the median cut 2 standard deviations, or severe malnutrition when it was below the median cut 3 standard deviations. PN duration was defined from the day receiving PN to the day of weaning off from PN or the death of the patient, as previously described [12]. All three IF patients were carrying stomas, and biopsies were obtained at the time of jejunoileal anastomosis and ileostomy, ileo-colonic anastomosis, or stoma closure surgery. Specimens were cut into full-thickness sections for immediate storage in (1) 10% buffered formalin for paraffin-embedded blocks and sections, (2) 10×genomics buffer for single-cell RNA-sequencing, and (3) DMEM/F12 culture medium for the construction of patient-derived organoids (PDOs). Clinical sample collection was approved by Ethics Committee of Xinhua hospital affiliated to Shanghai Jiao Tong University School of Medicine (XHEC-C-2021-110-1).

### 2.2. 10× Genomics Single-Cell RNA Sequencing (scRNA-seq)

#### 2.2.1. Preparation of Single-Cell Suspension

Specimens were dissociated into single cells using dissociation solution (0.35% collagenase IV, 2 mg/mL papain, 120 U/mL DNase I) and the resultant cell suspension was filtered through a 30–70 μm filter. After centrifugation for 5 min (4 °C, 300× *g*), the cell pellet was resuspended in 100 ul of PBS containing 0.04% BSA. The remaining erythrocytes were then lysed in 1 mL of erythrocyte lysis buffer (MACS 130-094-183) at room temperature for 10 min. Cell viability was examined with trypan blue assay, and single-cell counts were adjusted to 700–1200 cells/mL.

#### 2.2.2. Chromium 10× Genomics Library and Sequencing

Briefly, 5000 single cells were captured using the 10× Genomics Chromium Single-Cell 30 kit (V3), followed by cDNA amplification and library construction according to standard procedures. Sequencing was performed using an Illumina NovaSeq 6000 sequencing system (paired-end multiplex run, 150 bp) by LC-Bio Technology Co. Ltd. (Hangzhou, China) with a minimum depth of 20,000 reads per cell.

#### 2.2.3. Identification of the Major Cell Types

Seurat software was used for dimensionality reduction and cell type identification. Briefly, dimensionality reduction was performed using PCA (principal component analysis) based on the first 2000 most variable genes. Cells were then clustered into different cell populations with a resolution of 0.8, and the results of clustering were visualized by t-distributed Stochastic Neighbor Embedding (tSNE). Marker genes were identified with the Wilcoxon rank-sum test with default parameters.

#### 2.2.4. Pathway Enrichment Analysis

Bioinformatic analysis was performed using Gene Ontology (GO) term classification (david.abcc.ncifcrf.gov (accessed on 1 December 2021)) and protein–protein interaction (PPI) networks (v10, string-db.org). Differentially expressed genes were defined as those genes which are expressed in >10% of the cells in a cluster and have average log2 (fold change) >0.26 with adjusted *p*-values < 0.05.

### 2.3. Culture and Treatment of Patient-Derived Organoids (PDOs)

Isolation and culture of PDOs were identical to that described previously elsewhere [13]. Briefly, ileal crypts were extracted with 15 mmol/L EDTA for 45 min at 4 °C, and were then suspended with Matrigel (Corning Inc., Corning, NY, USA) and human intestinal organoid growth medium (Cat. #K2002-HI, bioGenous Human Intestinal Organoid Kit). After 2–3 passages of subculture, the PDOs were treated with 0.5 mM decanoic acid (DA) for 5 days, with the medium containing DA was changed every 2–3 days.

### 2.4. Cell Culture and Treatment

Caco-2 cells (American Type Culture Collection, Manassas, VA, USA) were cultured in DMEM supplemented with 10% FBS at 37 °C in a humidified atmosphere containing 5% CO2. Cells were exposed to 0.1%, 0.2%, and 0.5% soybean oil-derived lipid emulsion (SOLE) for 48 h; exposed to 0.2 mM, 0.5 mM, and 1 mM decanoic acid (DA) for 48 h; 50 μM etomoxir (ETO) was added 1 h prior to SOLE for inhibition of fatty acid oxidation.

### 2.5. TPN Rat Model

A total of 10 male Sprague Dawley specific-pathogen-free rats (3-week old; 80 ± 5 g) were maintained in an environment-controlled room (25 ± 0.5 °C, humidity 40–60%, 12-h light/dark cycle). After a 1-week acclimatization, rats were randomized to the sham or TPN group. Surgery procedure and composition of TPN solution were identical to those previously described [14]. All animals were sacrificed at 7 days after surgery, and the ileal tissues were isolated about 10 cm proximal to the ileocecal valve for histological examination. This study was approved by Xin Hua Hospital Animal Use Committee (XHEC-F-2020–008).

### 2.6. Immunofluorescence

For in vivo studies, tissue sections were de-paraffinized and rehydrated using standard techniques. Antigen retrieval was performed with 1 mM EDTA (pH = 9) buffer at 95 °C for 10 min. For whole-mount immunofluorescence staining, PDOs were incubated with rabbit anti-KLF4 (1:200) at 4 °C overnight. Images were captured and analyzed using Leica DMI6000B microscope coupled with Leica LAS X software (Leica, Germany).

### 2.7. Quantitative Real-Time Polymerase Chain Reaction (qPCR)

Total RNA was isolated using TRIzol total RNA isolation reagent (Thermo, Waltham, MA, USA), and the reverse transcription was performed using PrimeScript II 1st Strand cDNA Synthesis Kit (Takara, Japan). The qPCR analysis was performed on QuantStudio Dx real-time PCR system (Thermo, Waltham, MA, USA). Primers used in this study are listed as follows: KLF4 5′-CATCTCAAGGCACACCTGCGAA-3′ (Forward)/5′-TCGGTCGCATTTTTGGCACTGG-3′ (Reverse), SLC6A4 5′-TCACAGTGCTCGGTTACATGGC-3′ (Forward)/5′-GAAAGTGGACGCTGGCATGTTG-3′ (Reverse), SLC39A7 5′-GACCACAATGACTGTCCTGCTAC-3′ (Forward)/5′-GCTGTCAGTAGTTGCAGACGCA-3′ (Reverse), SLC51A 5′-TCTTCCTGGAGGATGCCGTCTA-3′ (Forward)/5′-TCCAGAGACCAAAGCAGCACAG-3′ (Reverse), SLC12A7 5′-CCTCAAGGATGCACAGAAGTCC-3′ (Forward)/5′-CGTAAGACCACGCCTTCAATGC-3′ (Reverse), β-actin 5′-CACCATTGGCAATGAGCGGTTC-3′ (Forward)/5′-AGGTCTTTGCGGATGTCCACGT-3′ (Reverse).

### 2.8. Western Blot

Protein lysates were extracted from PDO cultures with Cell Recovery Solution (Cat. #354253, Corning). The proteins (30 μg) were separated on NuPAGE 10% Bis-Tris gels (Thermo, Waltham, MA, USA) and transferred using the iBlot2 system (Thermo, Waltham, MA, USA). Representative protein bands were acquired using ChemiDoc™ Touch Imaging System (Bio-Rad, Hercules, CA, USA).

### 2.9. Reagents

Cell culture reagents for human organoids were purchased from BioGenous (Cat. #K2002-HI, Hangzhou, China). Matrigel (Cat. #356231) was purchased from Corning (Shanghai, China). Rabbit anti-KLF4 (Cat. #11880-1-AP) was purchased from Proteintech (Wuhan, China). All other chemicals were purchased from Sigma-Aldrich (Darmstadt, Germany).

### 2.10. Statistical Analysis

Data were statistically analyzed and plotted using GraphPad Prism 8.0. Variables were analyzed either by Student’s *t*-test for two groups or ANOVA analysis for multiple groups. A difference is considered significant when *p* < 0.05.

## 3. Results

### 3.1. Clinical Characteristics of IF Patients

The clinical characteristics of pediatric IF patients are summarized in Table 1. All the three controls had never received PN, while all the three IF patients were on PN. Nutritional diagnosis for control individuals was “Normal”, while that for IF patients was “Severe malnutrition”.

### 3.2. Decreased KLF4 and SLC Transporters in the Mature Enterocytes (mEC) of IF Patients

In total, 4668 epithelial cells from 3 control donors and 3717 epithelial cells from 3 IF donors were analyzed. Based on previously reported cell markers, five main known cell types were identified, including stem cells and enterocyte progenitors (SC/proEC), mature enterocytes (mEC), goblet cells (GC), Paneth cells (PC), and enteroendocrine cells (EEC). The number of cells in each cluster was similar in all groups except GC (Appendix A). Markers used for validation and projection of each cluster are shown (Appendix A), including Epithelial Cell Adhesion Molecule (EPCAM) to identify all the epithelial cells, lysozyme (LYZ) to identify PC, mucin 2 (MUC2) to identify GC, and chromogranin B (CHGB) to identify EEC. Olfactomedin 4 (OLFM4) and SRY-Box Transcription Factor 9 (SOX9) were used to identify SC/proEC. Alkaline Phosphatase-Intestinal (ALPI) and lactase (LCT) were used to identify mEC. Violin plots suggest that KLF4 was mainly enriched in the mEC and GC clusters (Figure 1A). In the mEC cluster, the expression of KLF4 evidently decreased in the IF patients compared to Controls (Figure 1B). Consistently, at the villus tip where mature enterocytes were enriched, in situ staining for KLF4 indicated robust KLF4 signal in Controls but not in IF patients (Figure 1C). Among all the differentially expressed genes, the top three enrichment categories for downregulated genes was “lipid metabolic process”, “transmembrane transporters”, and “proteolysis” (Appendix A), while the top three enrichment categories for upregulated genes was “detoxification of copper ion”, “negative regulation of growth”, and “cellular response to metal ion” (Appendix A). Notably, a number of SLC transporters were significantly decreased in IF patients, including SLC51A and SLC10A2 (transporting bile acids), SLC6A4/8/19 (transporting neurotransmitters), SLC3A1 and SLC7A7/9 (transporting amino acids), and SLC12A7 and SLC39A4/7 (transporting ions) (Figure 1D). Taken together, these results suggested that IF patients exhibited some features of functional defects in mature enterocytes, including the decreased expression of KLF4 and a variety of SLC transporters.

### 3.3. Characteristics of KLF4+ mEC

In order to further validate the role of KLF4 in mature enterocytes, we analyzed the differential characteristics of KLF4+ and KLF4− mEC. A total of 83 genes, including KLF4, were identified as differentially expressed genes between KLF4+ and KLF4− mEC. Protein–protein interaction (PPI) analysis indicated that most of the interactions were enriched in gene regulation complexes and transporters (Figure 2A). Notably, a number of SLC transporters that were decreased in IF patients were also substantially downregulated in KLF4− mEC, including SLC6A4/8, SLC7A9, SLC39A7, SLC12A7, and SLC51A (Figure 2B). Moreover, in order to avoid the bias caused by small sample size, we then evaluated the correlation of KLF4 with these SLC transporters by extracting data from public databases (GTEx). As shown, the expression of KLF4 exhibited significant correlation with SLC6A4/8, SLC7A9, SLC39A7, SLC12A7, and SLC51A (Appendix A). Taken together, these results suggested that KLF4− mEC exhibited lower levels of SLC transporters, which may cause functional defects in transporting certain substances.

### 3.4. Role of Enteral Nutrition in the Regulation of Intestinal KLF4 Expression

We then assumed that the loss of KLF4 in mEC was partially attributed to insufficient enteral nutrition. As shown in the control rats, robust KLF4 signal was detected at the villus tip where mature enterocytes were enriched. On the contrary, KLF4 expression was hardly detectable at the villus tip in TPN rats. Interestingly, however, robust signal was detected in the bottom of crypts where stem cells and progenitors were enriched (Figure 3). Given the hypothesis that the abnormal expression of KLF4 along the villus–crypt axis may be a result of the distinct routine of nutrient intake, we then examined the effect of varying nutrients on the expression of KLF4 in vitro. Notably, soybean oil-derived lipid emulsion (SOLE), which was used in the TPN solution, dose dependently (0.1–0.5%) induced the expression of KLF4 in Caco-2 cells (Appendix A). Moreover, SOLE-induced KLF4 was independent on fatty acid oxidation (FAO), as ETO (FAO inhibitor) did not alleviate the effect of SOLE on KLF4. These findings suggested that inducible KLF4 expression was likely attributed to the signal transduction of certain fatty acids instead of lipid metabolism (Appendix A). Finally, we found that decanoic acid (DA) was the key ingredient to induce KLF4. As shown, DA (0.2–1 mM) dose-dependently increased KLF4 expression (Appendix A), along with a number of SLC transporters, including SLC6A4, SLC12A7, SLC39A7, and SLC51A (Appendix A). Collectively, these results suggested that robust KLF4 expression may be dependent on stimulation of enteral nutrients, especially on fatty acids such as DA.

### 3.5. Effect of Decanoic Acid on the Expression of KLF4 in Patient-Derived Organoids (PDOs)

Intestinal organoids derived from two individual IF patients were then used to further consolidate the effect of DA on the regulation of KLF4 expression. Morphologically, no significant changes were observed after DA treatment, as the number of PDOs per well and organoid size remained unchanged (Figure 4A). At the mRNA level, DA increased the expression of KLF4 by 51.1%, along with SLC6A4 (increased by 89.7%) and SLC12A7 (increased by 22.3%) (Figure 4B). At the protein level, the expression of KLF4 increased by 98.7% upon DA treatment (Figure 4C). Consistently, immunofluorescence indicated that KLF4 was significantly increased by DA treatment (Figure 4D). Taken together, these results demonstrated that KLF4 within IF patient-derived organoids could be evidently induced by DA treatment in vitro.

## 4. Discussion

The incidence of the leading cause of pediatric IF, necrotizing enterocolitis (NEC), has remained relatively stable over the previous decades despite advances in the care of the neonate. In the United States, the mean incidence of NEC was 3–11% in 1997–2000 and 5–15% in 2003–2007 [17,18]. Due to septic episodes, metabolic disturbances, recurrent hospitalizations, and a poor quality of life, the care of children with IF remains a challenging issue even in developed countries. Therefore, a better knowledge of the pathogenesis of IF and new strategies for treating IF patients is urgently needed.

The overall goal in treating IF is to achieve enteral autonomy through intestinal adaptation so as to shorten PN duration and to prevent the associated complications, such as intestinal failure-associated liver disease (IFALD) [19]. In animal models, a wealth of functional and morphometric adaptive responses has been described, including increased cellular proliferation and angiogenesis that may stimulate mucosal growth and enhance absorptive capacities [20,21]. Additionally, some animal studies have also shown that intestinal adaptation may be associated with the increased expression of transporters critically important in nutrient, electrolyte, and water absorption (e.g., sodium glucose, Na/H exchangers), even independently of the increase in enterocyte mass [22,23]. In humans however, gross changes in intestinal morphological features (e.g., small bowel lengthening and dilation) have been shown [24,25], although only limited evidence is available regarding modification of intestinal microarchitecture and changes in transcriptomic regulation.

One of the key findings in this study is the identification of KLF4 as the hub gene responsible for the functional deficit of mature enterocytes in IF patients. The KLF members are a family of phylogenetically conserved, broadly expressed transcription factors initially identified in Drosophila melanogaster [26]. Each of the current 18 members contains three C2H2-type zinc finger motifs, which recognize GC-rich DNA binding sequences and thereby transactivate or repress target genes involved in diverse physiological and pathological processes [27]. Among the KLF members, KLF4 is particularly critical in the maintenance of intestinal epithelial homeostasis. The expression of KLF4 occurs in a gradient along the intestinal crypt–villus axis with the highest expression at the villus tip, which correlated strongly with an increasing degree of differentiation. By using the Vil-Cre system, Ghaleb et al. found that the conditional ablation of KLF4 led to the failure of Goblet cell differentiation, which highlighted the role of KLF4 in maintaining intestinal epithelial morphology [28]. Moreover, Tianxin et al. demonstrated that KLF4 may also play a central role in regulating the proliferation of stem cells and/or tuft cells, and in regulating the maturation of Paneth cells through crosstalk with Wnt signaling [29]. Additionally, KLF4 has also been reported to regulate the expression of multiple nutrient transporters, including zinc, folate, and biotin [30,31,32]. Interestingly, by using scRNA-seq in this study, we found significantly decreased expression of KLF4 in the mature enterocytes of IF patients, concomitant with decreased expression of SLC transporters. Indeed, in the gastrointestinal tract both ATP-binding cassette transporters (ABC transporters) and solute carrier (SLC) transporters play a pronounced role in the absorption and elimination of a broad range of nutrients, drugs, toxins, and their derivatives. However, SLC transporters do not require ATP, and thus play a more important role in transporting the nutrients along their concentration gradient, thereby facilitating the utilization of enteral nutrition. For example, SLC7A subfamily members belong to a family of amino acid transporters, which play a role in the high-affinity and sodium-independent transport of cystine and neutral and dibasic amino acids [33]. Therefore, the decreased expression of intestinal SLC7A9 may contribute to the reduced efficiency of protein digestion in IF patients. In addition, SLC51A serves as one of the intestinal basolateral transporters responsible for bile acid and steroid (e.g., estrone 3-sulfate) export from enterocytes into portal blood, therefore playing a role in the enterohepatic circulation [34]. Interestingly, some of the SLC6A subfamily members were also found suppressed in IF patients, which are responsible for the transport of neurotransmitters [35]. As a result, we speculated that these dysregulated transporters may not necessarily contribute to the malabsorption of nutrients in pediatric IF patients; instead, they may have affected the uptake and release of microbial-derived signaling neurotransmitters (e.g., dopamine, serotonin, and indoles), thereby affecting cognition, emotion, stress resilience and recovery, appetite, and metabolic balance through the gut–brain axis. Despite the bioinformatics evidence showing the strong correlation between KLF4 and those SLC transporters mentioned above (Figure 1C, Figure 2B and Appendix A), one of the limitations in this study is that we did not fully verify the expression of these SLC transporters as KLF4 downstream targets in vitro, and thus more studies are required in future for further validation.

Another one of the key findings in this study is the potential cause of reduced KLF4 expression in IF patients. By using a rodent model of TPN, we demonstrated that the deprivation of enteral nutrition was sufficient to suppress the KLF4 expression at the villus tip (Figure 3). This was reasonable as the luminal nutrients play an important role in the maintenance of intestinal homeostasis either by stimulating mucosal hyperplasia through direct contact with epithelial cells or by stimulating the secretion of trophic gastrointestinal hormones [36,37]. More importantly, however, peripherally administered TPN solution containing SOLE dramatically induced the KLF4 expression in the bottom of crypts, which implied a possibility that some of the nutrients incorporated in the TPN solution may be able to induce KLF4 expression. In the preliminary study, nutrients including various fatty acids have been tested in vitro for their effect on the regulation of KLF4 by using Caco-2 cells, and finally we found that decanoic acid (DA), a key component of medium-chain triglycerides (MCT), exhibited significant effect on KLF4 regulation (Figure 4). For decades, the role of MCT in the intestinal adaptive process of IF patients has remained controversial. Theoretically, the absorption process of MCT would increase fat uptake and provide a faster energy supply, as they can be directly absorbed across the enterocytes into the portal circulation. A study in patients with an intact colon found that a diet containing MCT improved fat absorption, which may be beneficial for the patients with bile acid or pancreatic insufficiency [38]. In contrast, a clinical trial in patients with jejuno- or ileostomy showed that high concentrations of MCT can cause osmotic diarrhea due to rapid hydrolysis of MCT [39]. To date, there are few studies using DA alone and thus the specific role of DA in intestinal homeostasis remains unclear. However, a group of researchers has recently demonstrated that the supplementation of DA can improve intestinal barrier function and antioxidation function via activating the G protein-coupled receptor-43 [40]. One of the limitations of this study is that we did not verify the effect of supplementation of DA on the KLF4 expression and enterocyte maturation in animal models of intestinal failure, and thus more in vivo studies are required in future to address this issue. Instead, IF patient-derived intestinal organoids were used to verify the effect of DA on KLF4 in vitro, since emerging evidence has recently demonstrated that the development of patient-derived organoids may potentially create a novel way of experimentally investigating gastrointestinal diseases, including Hirschsprung’s disease and short bowel syndrome [41]. Moreover, these human-derived organoids have been demonstrated to be feasible for generating a tissue-engineered small intestine (TESI) with the presence of major differentiated epithelial cell types of mature human small intestine (e.g., enterocytes, enteroendocrine cells, Goblet cells, and Paneth cells) [42], which highlighted the development of autologous tissue transplantation as a treatment for IF in future.

## 5. Conclusions

In summary, the decreased expression of KLF4 and SLC transporters in mature enterocytes may be responsible for nutrient malabsorption in pediatric IF patients. Decreased expression of KLF4 may be attributed to insufficient enteral nutrition. The supplementation of decanoic acid may be a potential strategy to promote cell maturation and functional improvement in enterocytes by targeting KLF4.

## Figures and Tables

**Figure 1 nutrients-15-02660-f001:**
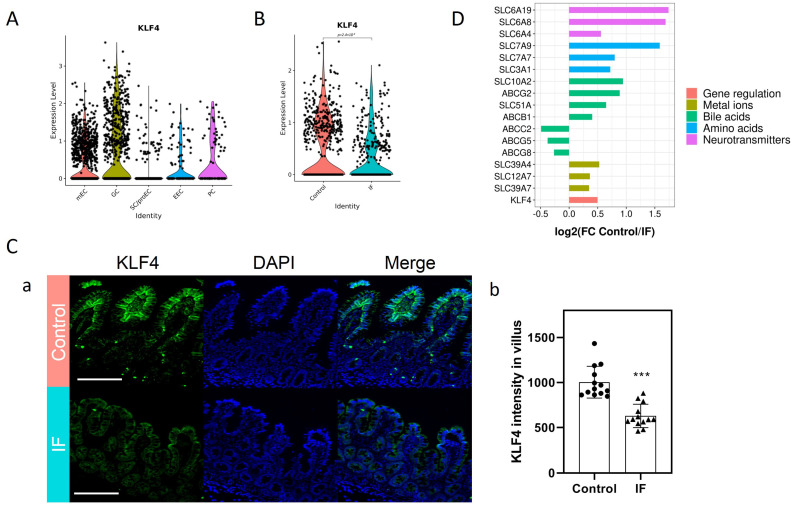
Expression of KLF4 in pediatric IF patients: (**A**) Violin plots showing KLF4 expression in each cluster; (**B**) violin plots showing KLF4 expression in mEC; (**C**) representative images (**a**) of KLF4 staining. Scale bar = 100 μm. (**b**) Quantification of KLF4 intensity. Data are presented as mean ± SD. *** *p* < 0.001. Three experiments were performed that showed similar results. (**D**) Expression pattern of KLF4 and transporters. Data are presented as Log2 (FC Control/IF). IF, intestinal failure.

**Figure 2 nutrients-15-02660-f002:**
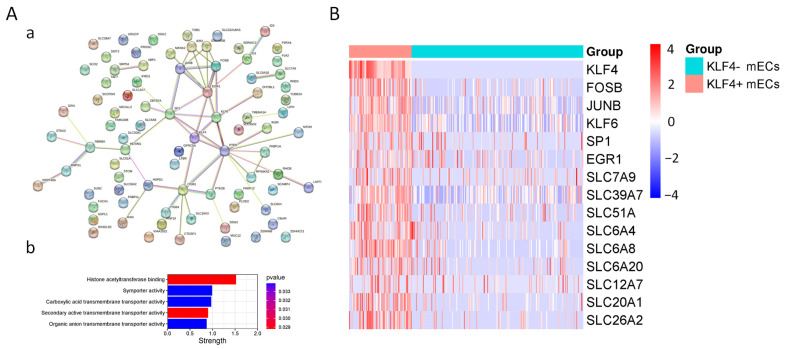
Characteristics of KLF4+ mECs. (**A**) Protein–protein interaction (PPI) showing the interaction of differentially expressed genes between KLF4+ and KLF4− mECs (**a**) and quantification for interaction strength (**b**). (**B**) Heatmap of differential gene expression signature in KLF4+ mECs.

**Figure 3 nutrients-15-02660-f003:**
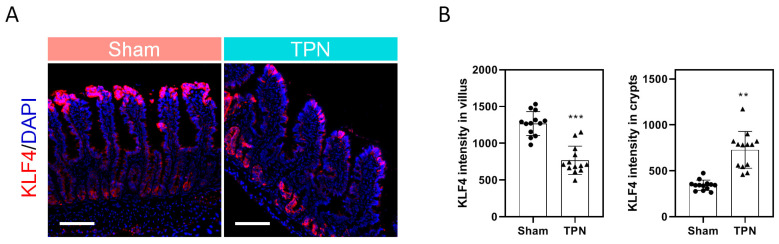
Role of enteral nutrients in the regulation of KLF4 expression (**A**) Representative images showing KLF4 expression in the villus of TPN rats. Scale bar = 100 μm. Three experiments were performed that showed similar results. (**B**) Quantification of KLF4 intensity. Data are presented as mean ±SD. ** *p* < 0.01, *** *p* < 0.001 (*n* = 5/group). TPN, total parenteral nutrition.

**Figure 4 nutrients-15-02660-f004:**
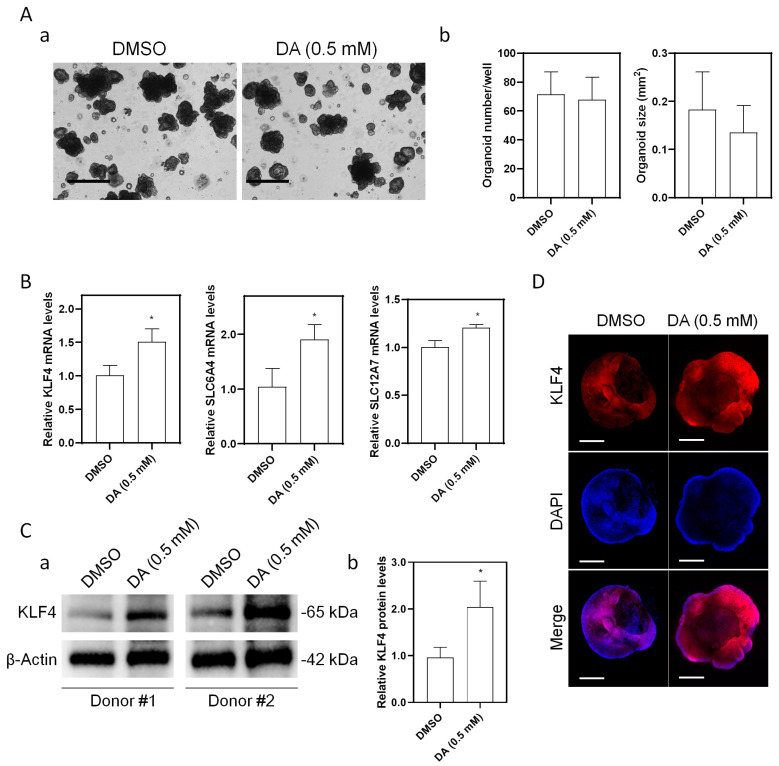
Effect of DA on KLF4 expression in PDOs. (**A**) Morphological changes of PDOs by DA. Scale bar = 1 mm. Data are presented as mean ± SD. * *p* < 0.05. (**B**) mRNA levels of KLF4, SLC6A4, and SLC12A7. Data are presented as mean ± SD. * *p* < 0.05. (**C**) Representative immunoblots of KLF4 (**a**) and quantification of fold change (**b**). Data are presented as mean ± SD. * *p* < 0.05. (**D**) Representative images showing inducible KLF4 expression (red) by DA treatment. Scale bar = 100 μm. Three experiments were performed that showed similar results. DA, decanoic acid.

**Table 1 nutrients-15-02660-t001:** Clinical characteristics of IF patients.

Group	Control	Control	Control	IF	IF	IF
ID	Patient #1	Patient #2	Patient #3	Patient #4	Patient #5	Patient #6
Gender	Female	Female	Female	Male	Male	Female
Age (years)	3	1	5	2	1	3
Weight (kg)	14	9.6	21	7.2	5.3	9.2
Gestational age (weeks)	38	37	37	41	29	38
Clinical diagnosis	Ileal duplication	Abdominal cyst	Ileal duplication	Short bowel syndrome	Short bowel syndrome	Short bowel syndrome
Residual small bowel length (cm)	/	/	/	149	108	117
“Normal” small bowel length values (cm) *	/	/	/	345.5	283.9	366.7
Remaining bowel anatomy	/	/	/	Partial colectomy, ileocecal valve preserved	Partial colecto-my, ileocecal valve preserved	Partial colecto-my, ileocecal valve not preserved
Months from the 1st resection	/	/	/	9	8	20
Weight for age	<M + 1 SD	<M + 1 SD	<M + 2 SD	<M – 3 SD	<M – 3 SD	<M – 3 SD
Nutritional diagnosis	Normal	Normal	Normal	Severe malnutrition	Severe malnutrition	Severe malnutrition
On/Off PN	/	/	/	On	On	On
Calories provided by PN (kCal/day)	/	/	/	492	470	639
Calories provided by EN (kCal/day)	/	/	/	241	160	200
Energy requirements (kcal/day) for parenteral nutrition **	/	/	/	468–540	397.5–450.5	598–690
Duration of PN (days)	/	/	/	166	123	230
Plasma ALT (U/L)	11	10	36	85.7	14.8	45
Plasma AST (U/L)	32	33.4	29	102	36.1	110.4
Plasma total bilirubin (μmol/L)	5.5	4	3.6	25.5	15.3	7.2
Plasma direct bilirubin (μmol/L)	1.5	0	<1	0	0	0
Albumin (g/L)	39.8	20.7	51.3	34	29.3	40.2

Normal levels of albumin (35–50 g/L); * Reprinted from ref. [15], ** Reprinted from ref. [16].

## Data Availability

The data presented in this study are available on request from the corresponding author.

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
