# Peer review of "Decreased Expression of KLF4 Leading to Functional Deficit in Pediatric Patients with Intestinal Failure and Potential Therapeutic Strategy Using Decanoic Acid"

_nutrients, 2023, doi:10.3390/nu15122660_

Round 1
Reviewer 1 Report
Quite interesting approach in the search of explaining the mechanisms behind intestinal adaptation in pediatric short bowel syndrome. Nevertheless, there are major questions that need to be answered by the authors:
1. The definition of If is not completely correct, because it is not because of the intolerance of enteral nutrition but because of the insufficiency to provided enough amounts of fluids and nutrients to cover all requirements. Please, refer to the definition by ESPEN: Intestinal failure (IF) is defined as the reduction of gut function below the minimum necessary for the absorption of macronutrients and/or water and electrolytes, such that intravenous supple- mentation is required to maintain health and/or growth.
2. Its is more appropriate to defined CIPO as a motility disorder rather than a neuromuscular disease.
3. Please, provide references for the paragraph 64-75.
4. Material and methods
a. Please, define clearly who were the controls and how they were chosen.
b. It is not clear when the biopsies were taken. Were the patients carrying and stoma and the biopsies were taken when the stoma was taken off? If the biopsies were taken just after the first resection it was not time enough to enhance adaptation. This is a clue point. Were they taken when the patients was receiving any kind of enteral support or were they exclusively TPN fed?
c. The approval from the IRB for the study in babies was not included. The one for the animal project was correctly quoted.
5. Results
a. The patients were ON PN (not wean on PN)
b. Crude data on BMI is not the right way when comparing patients with different ages: SD or Z score are more appropriate.
c. In M&M section cut off criteria for malnutrition or normal nutrition were not provided.
d. Normal levels of albumin were not provided. Control # 2 has a quite low albumin level. The reason?
e. No data on type of nutrition support was provided in controls and only partial in patients: calories in PN, EN complementary support, etc.
Without all these crucial information it is hard to interpret the results from the complex study on intestinal cells population and gene expression. So I will not proceed further in the analysis without an answer for all these questions.
It is OK
Reviewer 2 Report
The authors investigate the role of Krüppel like Factor 4 (KLF-4) in children with intestinal failure. They first looked at the KLF-4 expression in pediatric patients with short bowel compared to normally nourished children and found decreased expression in the villus of the 3 children. They next found decreased expression of SLC transporters in KLF-4 minus (KLF-4-) mature Enterocytes (mEC) compared to KLF-4+ mEC. They concluded that KLF-4 regulates SLC transporters. In the next step they looked for KLF-4 expression in rats on TPN and found decreased expression in the tip of the villus but increased expression in crypts. Using Caco-2 cells they showed that addition of decanoid acid (DA) induces KLF-4 expression. Using patient derived organoids (PDO), DA induces KLF-4 together with SLC expression at the RNA as well as the protein level.
The authors conclude that KLF-4 is a hub gene in regulating EC maturation and function and might therefore be a potential point of regulation using DA as “inducer” of KLF-4. It remains however unclear whether the role of DA in regulating KLF-4 is via enteral or parenteral route. Using the severely malnourished children with short bowel as model of KLF-4 down-regulation, it remains unclear whether these children were on TPN only or also on enteral nutrition or both? Patient characteristics in Table 1 should therefore be precised specially how much bowel length has been lost or remain in situ. Furthermore, the authors showed that TPN treated rats express less KLF-4 in the villus tip. However, TPN induces severe villus atrophy as shown by the authors themselves. Is the effect on TPN a loss of KLF-4 or a loss of villus ? Unfortunately, as the authors stated themselves in the discussion (line 357-358), the did not reexpose rats with DA. Furthermore, it would have been interesting to see whether reexposition of rats after 1 week of TPN with enteral nutrition leads to induction of KLF-4 expression in the villus tip. Instead, they induced KLF-4 by DA in PDOs. Whether the in vitro culture of Caco-2 cells reflects the enteral or the parenteral route remains undiscussed.
Minor comments:
1. In all figures the number of experiments performed should be stated.
2. Fig. S4b: what concentration of SOLE was used?
3. 10 rats were equally randomized into 2 groups (TPN and oral nutrition, 5 per group) In Figure 3, results show 13 points per group suggesting that more animals have been used. Please clarify. The same for Figure 1.
4. In the second part of the experiments, KLF-4 + and KLF-4 mature EC were analyzed. How were the cells generated?
Minor cahnges such as "did'nt" should be changed to "did not".
Reviewer 3 Report
Thanks for this interesting study. I am not qualified to review all the technical aspects of the molecular biology research here. However having significant clinical experience and clinical study background, I wanted to raise a few topics with the authors.
My main concern is the scarce background data on the IF patients bowel anatomy. It makes a huge impact to have remaining 15% of SB and 50% colon vs. end-jejunosotmy vs. 10% SB with ICV and colon intact. I suggest to add the bowel anatomy data to Table1.
-Also, knowledge on parenteral vs enteral nutrition provision is essential when it comes to the stage of enteral adaptation. Add to table1
-All three IF pts were severely malnourished despite PN, which is different from the average IF therapy aiming to provide sufficient energy for maintenance and growth, in combination of PN&EN.
Thus the main body of information comes from the rat study. Page 7 discusses the role of fatty acids especially DA. The role of SOLE is somewhat obscure - do the authors suggest SOLE in PN affects enterocyte function?
Especially Results section requires significant editing. Now there are full stops missing, misspellings etc.
Round 2
Reviewer 1 Report
See in the attached file

Reviewer 2 Report
Comments have been answered appropirately.
Author Response
We highly appreciate the valuable comments you have raised, which are so thought-provoking and important for us to improve the academic quality of this work. Thank you!
Reviewer 3 Report
Thank you for your responses.
In Table 1, I suggest to correlate the SB centimeters to known ”normal” SB length values presented in (Struijs MC, Diamond IR, de Silva N, Wales PW. Establishing norms for intestinal length in children. J Pediatr Surg. 2009 May;44(5):933-8. doi: 10.1016/j.jpedsurg.2009.01.031. PMID: 19433173), and also clearly state remaining bowel anatomy (colon present? end-jejunostomy?)
Also, in children daily energy provision should be correlated to child weight, as daily needs are dependent on child size (not age). If under 10 kg, daily kcal/kg may suffice (see Joosten K, Embleton N, Yan W, Senterre T; ESPGHAN/ESPEN/ESPR/CSPEN working group on pediatric parenteral nutrition. ESPGHAN/ESPEN/ESPR/CSPEN guidelines on pediatric parenteral nutrition: Energy. Clin Nutr. 2018 Dec;37(6 Pt B):2309-2314. doi: 10.1016/j.clnu.2018.06.944. Epub 2018 Jun 18. PMID: 30078715.)
Regarding the three PN dependent patients, I’m still unsure if they present on optimal example, as even children dependent on total PN still should not be so undernourished in modern practice. This leads to question if here the findings associated with gut mucosa may not the the primary cause of ongoing PN dependency but rather secondary to ongoing malnutrition.
Requires some editing
